# Enhancing Conditional Risk Control in Image Segmentation with Adaptive Conformal Prediction

## Abstract

Uncertainty quantification is crucial in high-stakes image segmentation, yet existing conformal risk control methods often exhibit highly variable conditional risk: some images suffer extreme false negative rates while others show minimal errors. We introduce Conformal Risk Adaptation (CRA), a framework that employs a novel score function, inspired by adaptive prediction sets, to create image-specific uncertainty regions. We formalize the risk control problem as a weighted quantile estimation task, which enables a computationally efficient, grid-search-free algorithm for threshold calculation. To ensure the reliability of our adaptive score function, we integrate a specialized non-parametric calibration method that enhances pixel-wise probability estimates. Experiments on polyp and crack segmentation demonstrate that CRA maintains valid marginal risk guarantees while delivering substantially more consistent conditional risk control across diverse images. This advancement provides practitioners with a principled approach to uncertainty quantification that adapts to individual cases while maintaining rigorous statistical guarantees, which is critical for personalized medical applications. We have provided code with implementation details in the repository below: `https://anonymous.4open.science/r/conformal-risk-adaptation-3BB2`.

## 1 Introduction

Image segmentation is a fundamental computer vision task with critical applications in medical diagnostics, autonomous driving, and structural health monitoring. While deep learning has significantly advanced segmentation performance, reliable uncertainty quantification remains challenging but essential for safety-critical applications. Traditional evaluation metrics like Dice or IoU provide overall performance measures but fail to offer instance-wise reliability guarantees.

Conformal prediction (CP) has emerged as a powerful framework for providing distribution-free uncertainty quantification with finite-sample guarantees. It constructs prediction regions that contain the true label with a user-specified probability, regardless of the underlying data distribution. Recent work on conformal risk control (CRC) Angelopoulos et al. (2024) has extended this framework to handle more complex performance metrics beyond simple miscoverage, including controlling the false negative rate in segmentation tasks.

However, applying CRC directly to image segmentation reveals a critical limitation. The primary issue is **weak conditional risk control**: While CRC guarantees risk control *on average* across a dataset, its performance on individual images can be erratic. A model might produce near-perfect segmentations for some images while making substantial errors on others. This problem is often exacerbated because the underlying confidence estimates from segmentation models can be unreliable. A fixed probability threshold, as used in standard CRC, thus becomes inadequate for nuanced, image-specific confidence levels, resulting in suboptimal prediction sets and inconsistent performance.

To overcome these challenges, we introduce Conformal Risk Adaptation (CRA), a comprehensive framework designed to deliver robust conditional risk guarantees. Our main contributions are:

- **Weighted Quantile Formulation**: We formalize the risk control problem as a weighted quantile estimation problem. This perspective not only provides theoretical clarity but, more importantly,

enables a highly efficient, grid-search-free algorithm for computing the exact risk-controlling threshold, a practical improvement over prior grid-search-based methods.

- **Adaptive Score Function**: The core of our framework is the CRA score function, which adapts the principles of Adaptive Prediction Sets (APS) from classification to the segmentation domain. This allows prediction sets to resize dynamically based on each image's unique confidence profile, directly addressing and mitigating conditional risk variability in segmentation.

- **Two-Stage Calibration**: To ensure the fidelity of our adaptive score, we integrate a two-stage calibration process, combining non-parametric probability recalibration with a stratified strategy inspired by group conditional conformal prediction. This approach enhances the reliability of the underlying probability estimates and tailors risk control to distinct groups of images.

Our comprehensive experiments on both medical and engineering image segmentation tasks demonstrate that our method consistently provides valid marginal risk control with significantly improved conditional performances compared to existing approaches, highlighting the practical advantages of our integrated calibration and adaptive scoring methodology.

## 2 PRELIMINARIES

### 2.1 PROBLEM SETUP

We consider the image segmentation task where each input image $X_i$ is associated with a ground truth segmentation mask $Y_i \subset \{1, 2, \ldots, N_i\}$, where $N_i$ represents the total number of pixels in image $X_i$. Each pixel is indexed by $j \in \{1, 2, \ldots, N_i\}$. Our goal is to construct a prediction set $\hat{C}(X_i) \subset \{1, 2, \ldots, N_i\}$ that controls the false negative rate in expectation, ensuring:

$$\mathbb{E}\left[1 - \frac{\left|\hat{C}(X_i) \cap Y_i\right|}{|Y_i|}\right] \leq \alpha, \tag{1}$$

where $\alpha \in (0, 1)$ is a user-specified risk level. The expectation in Equation 1 reflects the average performance of the prediction set $\hat{C}(X_i)$ over random draws of the test data. This metric represents the proportion of true positive pixels that are incorrectly excluded in the prediction set, which is particularly important in applications like medical imaging where missing regions of interest can have serious consequences.

### 2.2 CONFORMAL RISK CONTROL

We briefly review the standard Conformal Risk Control (CRC) framework Angelopoulos et al. (2024), which extends conformal prediction to control the expected value of any monotone loss function. For segmentation, CRC controls false negative rate by applying a threshold to the pixel-wise probabilities produced by a base model. The optimal threshold value is determined through a calibration procedure on a held-out calibration dataset.

Given a prediction model $\hat{p}(X_i) = (\hat{p}_1(X_i), \ldots, \hat{p}_{N_i}(X_i))$, where $\hat{p}_j(X_i)$ estimates $\mathbb{P}(j \in Y_i | X_i)$ for pixel $j$, the CRC approach proceeds through the following steps:

1. **Score Definition**: Define conformity scores $s(X_i, j) = \hat{p}_j(X_i)$ for each pixel $j$ in image $X_i$.

2. **Threshold Calibration**: Compute the calibrated threshold $\tau$ using a held-out calibration dataset $\mathcal{I}_{\text{cal}}$ that controls the expected false negative rate at level $\alpha$.

3. **Prediction Set Construction**: For a new test image, construct prediction sets $\hat{C}(X_i, \tau) = \{j : s(X_i, j) \geq 1 - \tau\}$ using the calibrated threshold.

This approach guarantees that $\mathbb{E}[1 - |\hat{C}(X_i, \tau) \cap Y_i|/|Y_i|] \leq \alpha$ over the data distribution, providing a distribution-free control of the false negative rate.

# 3 METHODOLOGY

## 3.1 WEIGHTED QUANTILE FORMULATION FOR RISK CONTROL

We improve the threshold calibration process as described in step 2 by connecting conformal risk control to weighted quantile estimation, enabling exact computation without expensive grid searches or precision limitations.

For each pixel $j$ in image $X_i$, the conformity score (CRC, as described in step 1) that measures how likely a pixel is to be included in the prediction set:

$$s(X_i, j) = \hat{p}_j(X_i), \tag{2}$$

where $\hat{p}_j(X_i)$ estimates the probability that pixel $j$ belongs to the target class.

With this definition, the calibrated threshold $\tau$ for controlling the false negative rate is defined as:

$$\tau = \inf \left\{ \tau' : \frac{n}{n+1} \sum_{i \in \mathcal{I}_{\text{cal}}} \sum_{j \in Y_i} \frac{\mathbb{1}\{s(X_i, j) < 1 - \tau'\}}{|Y_i|} + \frac{B}{n+1} \leq \alpha \right\}, \tag{3}$$

where $n$ is the size of the calibration set, $B$ is the upper bound of the loss function, and $\alpha$ is the desired risk level.

This formulation can be interpreted as finding the $[(n+1)\alpha - B]/n$-th quantile of the weighted distribution of scores $\{(s(X_i, j), 1/|Y_i|) : i \in \mathcal{I}_{\text{cal}}, j \in Y_i\}$, where $1/|Y_i|$ serves as the weight for each score $s(X_i, j)$. This weighted quantile formulation applies to any valid score function, including the proposed CRA as defined in the following subsection.

**Theorem 1.** *For any score function $s(X_i, j)$ that ranks pixels based on their likelihood of inclusion, the optimal threshold $\tau$ that controls the expected false negative rate at level $\alpha$ can be computed as the solution to the weighted quantile formulation in Equation 3.*

*Proof.* For a new test point $(X_{n+1}, Y_{n+1})$, we define the false negative rate as:

$$1 - \frac{|\hat{C}(X_{n+1}, \tau') \cap Y_{n+1}|}{|Y_{n+1}|},$$

where $\hat{C}(X_{n+1}, \tau') = \{j : s(X_{n+1}, j) \geq 1 - \tau'\}$ is the prediction set with threshold $\tau'$.

Our goal is to find a threshold $\tau$ such that $\mathbb{E}\left[1 - \frac{|\hat{C}(X_{n+1}, \tau) \cap Y_{n+1}|}{|Y_{n+1}|}\right] \leq \alpha$.

Following the CRC framework, if we have calibration data $(X_i, Y_i)_{i=1}^n$ and use the risk estimator:

$$\hat{R}(\tau') = \frac{1}{n} \sum_{i=1}^n \left( 1 - \frac{|\hat{C}(X_i, \tau') \cap Y_i|}{|Y_i|} \right),$$

then choosing $\tau$ as:

$$\tau = \inf \left\{ \tau' : \frac{n}{n+1} \hat{R}(\tau') + \frac{B}{n+1} \leq \alpha \right\},$$

guarantees that $\mathbb{E}\left[1 - \frac{|\hat{C}(X_{n+1}, \tau) \cap Y_{n+1}|}{|Y_{n+1}|}\right] \leq \alpha$.

For our score function $s(X_i, j) = \hat{p}_j(X_i)$, we have:

$$1 - \frac{|\hat{C}(X_i, \tau') \cap Y_i|}{|Y_i|} = 1 - \frac{|\{j \in Y_i : s(X_i, j) \geq 1 - \tau'\}|}{|Y_i|}$$

$$= \frac{|\{j \in Y_i : s(X_i, j) < 1 - \tau'\}|}{|Y_i|} = \sum_{j \in Y_i} \frac{\mathbb{1}\{s(X_i, j) < 1 - \tau'\}}{|Y_i|}.$$

Substituting this back, we get Equation 3. $\qquad\square$

## 3.2 CONFORMAL RISK ADAPTATION

We introduce Conformal Risk Adaptation (CRA), a novel score function designed specifically for segmentation. CRA adapts the core principle of Adaptive Prediction Sets (APS) Romano et al. (2020b), a technique from classification, to the segmentation domain. Unlike standard CRC which applies a uniform probability threshold across all images (as described in step 1), our CRA score allows the prediction set to dynamically resize based on the model's confidence for each specific image.

For each image $X_i$, we define an adaptive prediction set:

$$\hat{C}(X_i, \alpha') = \underset{C \subset \{1,...,N_i\}}{\arg\min} \ \{|C| : \sum_{j \in C} \hat{p}_j(X_i) \geq (1 - \alpha') \sum_{j=1}^{N_i} \hat{p}_j(X_i)\}, \qquad (4)$$

which selects the smallest set of pixels that captures at least $(1 - \alpha')$ of the total predicted probability mass. In case $\hat{C}(X_i, \alpha')$ has more than 1 element, i.e., more than 1 set has the same minimum number of elements, choose the one that has the largest summation of probabilities:

$$\underset{C_1 \in \hat{C}(X_i, \alpha')}{\arg\max} \sum_{j \in C_1} \hat{p}_j(X_i).$$

This is equivalent to sorting pixels by $\hat{p}_j$ in descending order and selecting the top-$k$ pixels whose probabilities sum to at least $(1 - \alpha') \sum_{j=1}^{N_i} \hat{p}_j(X_i)$. Similar to APS, CRA accounts for the relative ranking of pixel probabilities rather than applying uniform thresholds across all images, allowing for more adaptive prediction regions.

The intuition behind this approach relates to the expected coverage, where $\mathbb{E}[|C \cap Y_i|] = \sum_{j \in C} p_j(X_i)$ and $\mathbb{E}[|Y_i|] = \sum_{j \in [N_i]} p_j(X_i)$. By controlling the ratio of these expected values, we can effectively control the false negative rate.

For efficient computation, we define a conformity score that captures each pixel's position in this adaptive ranking. Unlike Adaptive Prediction Sets (APS) for classification, which typically use the predicted probability of the true class as a score, our CRA score for segmentation is normalized by the total predicted probability mass for the image:

$$s(X_i, j) = \frac{\sum_{j'=1}^{N_i} \hat{p}_{j'}(X_i) \mathbb{1}\{\hat{p}_{j'}(X_i) < \hat{p}_j(X_i)\}}{\sum_{j'=1}^{N_i} \hat{p}_{j'}(X_i)}. \qquad (5)$$

This normalization, by dividing by $\sum_{j'=1}^{N_i} \hat{p}_{j'}(X_i)$, makes the score image-specific and robust to variations in the overall confidence or "brightness" of an image's segmentation. It represents the proportion of the total predicted probability mass accumulated up to pixel $j$. This allows the prediction set to dynamically adapt to each image's unique confidence profile, directly addressing the issue of conditional risk variability. This adaptive score function replaces the simple probability-based score in step 1, while the threshold calibration (step 2) and prediction set construction (step 3) steps remain structurally similar but operate on our adaptive scores.

**Theorem 2.** *The adaptive prediction set $\hat{C}(X_i, \alpha')$ is equivalent to the set of pixels with conformity scores above a threshold: $\hat{C}(X_i, \alpha') = \{j : s(X_i, j) \geq 1 - \alpha'\}$.*

*Proof.* Let us define $S(X_i, 1 - \alpha') = \{j : s(X_i, j) \geq 1 - \alpha'\}$. We need to show that $\hat{C}(X_i, \alpha') = S(X_i, 1 - \alpha')$.

The conformity score $s(X_i, j)$ in Equation 5 represents the normalized cumulative probability mass of all pixels with probability strictly less than $\hat{p}_j(X_i)$. A score of $s(X_i, j) \geq 1 - \alpha'$ means that pixel $j$ is among the highest probability pixels whose cumulative probability mass accounts for at least $(1 - \alpha')$ of the total probability mass.

Let us examine what the condition $s(X_i, j) \geq 1 - \alpha'$ means in terms of probabilities:

$$\frac{\sum_{j'=1}^{N_i} \hat{p}_{j'}(X_i) \mathbb{1}\{\hat{p}_{j'}(X_i) < \hat{p}_j(X_i)\}}{\sum_{j'=1}^{N_i} \hat{p}_{j'}(X_i)} \geq 1 - \alpha'.$$

If we arrange all pixels in order of increasing probability and include pixel $j$ and all pixels with higher probabilities in our set $C$, then:

$$\frac{\sum_{j' \in C} \hat{p}_{j'}(X_i)}{\sum_{j'=1}^{N_i} \hat{p}_{j'}(X_i)} \geq 1 - \alpha',$$

which is equivalent to:

$$\sum_{j' \in C} \hat{p}_{j'}(X_i) \geq (1 - \alpha') \sum_{j'=1}^{N_i} \hat{p}_{j'}(X_i).$$

This is precisely the definition of $\hat{C}(X_i, \alpha')$ in Equation 4. By construction, the set $S(X_i, 1 - \alpha')$ contains exactly those pixels that satisfy the above inequality and is the smallest such set (as pixels are added in descending order of probability). Therefore, $\hat{C}(X_i, \alpha') = S(X_i, 1 - \alpha')$. $\qquad\square$

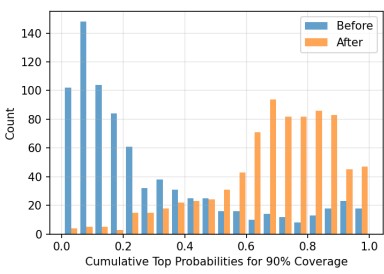 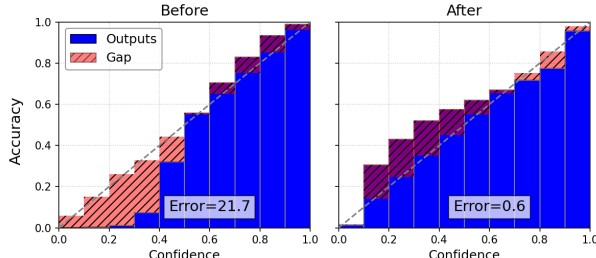

Figure 1: Left: Distribution of the image-specific probability mass proportion threshold (equivalent to $(1-\alpha')$ from Equation 4 for CRA scores) required to achieve $(1-\alpha)$ coverage for each image, before (blue) and after (orange) probability calibration. Right: Expected Calibration Error (ECE) before and after probability calibration, illustrating a substantial reduction in ECE from 21.7% (uncalibrated) to 0.6% (calibrated).

### 3.3 PROBABILITY CALIBRATION FOR ACCURATE PROBABILITY ESTIMATION

While CRC only relies on the relative ordering of pixel probabilities (meaning monotone calibration would not affect its performance), our CRA approach depends on accurate estimation of the total probability mass $\sum_{j=1}^{N_i} \hat{p}_j(X_i)$. Therefore, we introduce a probability calibration framework specifically designed for segmentation models.

Given predicted probabilities $\hat{p}_j(X_i)$ from a base segmentation model, we seek a calibration function $f : [0, 1] \to [0, 1]$ that satisfies:

- **Monotonicity**: For any $\hat{p}_a \leq \hat{p}_b$, we require $f(\hat{p}_a) \leq f(\hat{p}_b)$.

- **Probability Matching**: The calibrated probabilities should minimize the empirical cross-entropy loss:

$$\mathcal{L}_{\text{emp}}(f) = -\frac{1}{|\mathcal{I}_{\text{val}}|} \sum_{i \in \mathcal{I}_{\text{val}}} \sum_{j=1}^{N_i} \left[ y_{ij} \log f(\hat{p}_j(X_i)) + (1 - y_{ij}) \log(1 - f(\hat{p}_j(X_i))) \right], \quad (6)$$

where $y_{ij} \in \{0, 1\}$ indicates whether pixel $j$ in image $X_i$ belongs to the target class.

We apply $f$ to all predicted probabilities $\hat{p}_j(X_i)$ across all images and pixels, learning this function via an isotonic regression calibration function trained on a validation set that is separate from both the test data and the calibration data, which is critical to ensure that this recalibration step does not interfere with the subsequent conformal calibration process. As shown in Figure 1, our probability calibration greatly reduced the Expected Calibration Error (ECE) from 21.7% to a significantly lower 0.6%.

**Remark.** *While probability calibration improves the accuracy of individual pixel-wise probability estimates, it typically relies on monotonic transformations (e.g., isotonic regression or monotonic neural networks). Such transformations preserve the **ranking** of probabilities and thus do not alter the relative order of pixel confidences within an image. Consequently, probability calibration alone cannot fully account for variations in the **distribution** of confidence scores across different images, which can lead to inconsistent conditional risk. This inherent limitation is why the subsequent stratified calibration step is crucial: it adapts the risk-controlling threshold based on image-level characteristics, thereby addressing the remaining conditional risk inconsistencies that probability calibration cannot resolve.*

### 3.4 STRATIFIED CALIBRATION FOR IMPROVED CONDITIONAL RISK CONTROL

To further reduce the gap between the conditional risk and the desired control level, we propose a stratified calibration approach inspired by group conditional conformal prediction. This enhances the threshold calibration (step 2) by adapting thresholds to different groups of images.

We define a partitioning function $g : \mathcal{X} \rightarrow \{1, 2, ..., K\}$ that assigns each image to one of $K$ strata based on the sum of predicted probabilities across all pixels in the image, $\sum_{j=1}^{N_i} \hat{p}_j(X_i)$. This sum serves as a proxy for the model's overall confidence in its segmentation of the image.

For each stratum $k \in \{1, 2, ..., K\}$, we compute a stratum-specific calibrated threshold:

$$\alpha_k' = \inf \left\{ \alpha : \frac{n_k}{n_k + 1} \sum_{i \in \mathcal{I}_{\text{cal}} : g(X_i) = k} \left( 1 - \frac{|\hat{C}(X_i, \alpha) \cap Y_i|}{|Y_i|} \right) + \frac{B}{n_k + 1} \leq \alpha \right\}, \qquad (7)$$

where $n_k = |\{i \in \mathcal{I}_{\text{cal}} : g(X_i) = k\}|$ is the number of calibration samples in stratum $k$.

For a new test image $X_{n+1}$, we determine its stratum $k = g(X_{n+1})$ by calculating its total probability mass and apply the corresponding threshold $\alpha_k'$.

The stratified calibration approach provides valid risk control within each stratum:

$$\mathbb{E} \left[ 1 - \frac{|\hat{C}(X_i, \alpha_k') \cap Y_i|}{|Y_i|} \middle| g(X_i) = k \right] \leq \alpha. \qquad (8)$$

This approach improves conditional risk control by adapting the calibration thresholds to images with different levels of predicted probability mass, which is important in segmentation tasks where the model's confidence can vary significantly across different images. Images with similar total predicted probabilities tend to exhibit similar risk characteristics, allowing for more tailored risk control.

## 4 RELATED WORK

### 4.1 CONFORMAL RISK CONTROL

Conformal prediction Vovk et al. (2005) provides distribution-free uncertainty quantification with finite-sample guarantees. The split conformal prediction Lei et al. (2018) has gained popularity due to computational efficiency. Recent work has extended conformal prediction to control risk metrics beyond miscoverage Bates et al. (2021); Angelopoulos et al. (2024). Angelopoulos et al. Angelopoulos et al. (2024) introduced Conformal Risk Control (CRC). Teneggi et al. Teneggi et al. (2023; 2025) proposed grouping strategies for pixels and semantic-specific control. Bereska et al. Bereska et al. (2025) introduced Spatially-Aware Conformal Prediction. Other adaptive methods include Argaw et al. (2022); He et al. (2025); Blot et al. (2025); Wundram et al. (2024); Chen et al. (2025); Brunekreef et al. (2024).

### 4.2 CONDITIONAL CONFORMAL PREDICTION

Although conformal methods guarantee marginal coverage, achieving conditional coverage is generally impossible without distributional assumptions Vovk (2012); Foygel Barber et al. (2021). Many works aim to improve conditional validity by modifying calibration Lei & Wasserman (2014); Guan

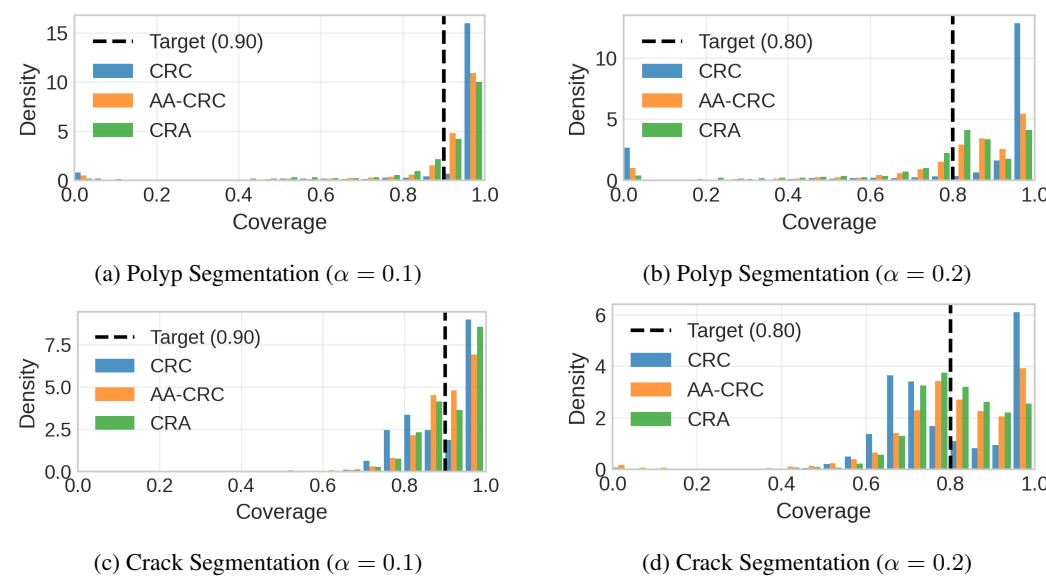

Figure 2: Coverage distribution comparison across different datasets and significance levels. Histograms show the empirical distribution of coverage rates for CRC (blue), AA-CRC (orange), and CRA (green) methods. The dashed vertical line indicates the target coverage rate of $(1 - \alpha)$. CRA produces more concentrated distributions centered around the target coverage rate.

(2023); Barber et al. (2023) or altering prediction rules Romano et al. (2019); Sesia & Romano (2021); Chernozhukov et al. (2021). Research on coverage under covariate shift includes Lei & Wasserman (2014); Tibshirani et al. (2019); Izbicki et al. (2022); Guan (2023); Hore & Barber (2023); Cauchois et al. (2024); Gibbs et al. (2025). Group conditional guarantees have been explored Toccaceli & Gammerman (2019); Gupta et al. (2020); Ding et al. (2023); Dunn et al. (2023); Kiyani et al. (2024b); Vovk et al. (2003); Romano et al. (2020a); Foygel Barber et al. (2021); Jung et al. (2023). Additional work includes learning features for conditional coverage Yuksekgonul et al. (2023); Kiyani et al. (2024a); Kaur et al. (2025); Bhattacharyya & Barber (2024); Prinzhorn et al. (2024); Silva-Rodr'ıguez et al. (2025). Our method builds upon these approaches by offering tighter finite-sample risk control guarantees without distributional assumptions.

## 5 EXPERIMENTS

### 5.1 EXPERIMENTAL SETUP

We evaluate three conformal prediction methods: CRC (Conformal Risk Control Angelopoulos et al. (2024)), AA-CRC (Automatically Adaptive CRC Blot et al. (2025)), and CRA (our proposed method). We evaluate on two segmentation tasks: polyp segmentation using datasets from ETIS, CVC-ClinicDB, CVC-ColonDB, EndoScene, and Kvasir with PraNet Fan et al. (2020), and crack segmentation using merged datasets Bianchi & Hebdon (2022) with DeeplabV3+ Chen et al. (2018).

For CRC and CRA, the evaluation data is split equally: 50% is designated for the calibration set ($\mathcal{I}_{\text{cal}}$) and 50% for the test set ($\mathcal{I}_{\text{test}}$). For AA-CRC, 25% of the data is allocated for training its threshold prediction model, 25% for calibration, and the remaining 50% for testing. All experiments were repeated over 100 random data splits to ensure statistical robustness.

We assess performance using four key evaluation metrics. While CRA primarily focuses on improving the Coverage Gap, we also provide Precision and Normalized Size for a complete assessment of the methods.

Marginal Coverage is given by $\frac{1}{|\mathcal{I}_{\text{test}}|} \sum_{i \in \mathcal{I}_{\text{test}}} \frac{|\hat{C}(X_i) \cap Y_i|}{|Y_i|}$, with a target of $1 - \alpha$. The Coverage Gap is defined as $\frac{1}{|\mathcal{I}_{\text{test}}|} \sum_{i \in \mathcal{I}_{\text{test}}} \left( \left| \frac{|\hat{C}(X_i) \cap Y_i|}{|Y_i|} - (1 - \alpha) \right| \right)$, where a lower value indicates better conditional

Table 1: Performance comparison of different methods across datasets and significance levels ($\alpha$). We report Marginal Coverage (target is $1 - \alpha$), Coverage Gap (fairness, lower is better $\downarrow$), Precision (higher is better $\uparrow$), and Normalized Size (lower is better $\downarrow$). Values are mean (std $\times \sqrt{399}$). **Bold** values indicate the best performance for each metric. A dagger ($\dagger$) on a Coverage Gap value for CRA indicates that the improvement over the second-best method is statistically significant (one-sided t-test, $p < 0.05$).

| $\alpha$ | Dataset | Method | Marginal Coverage | Coverage Gap $\downarrow$ | Precision $\uparrow$ | Size $\downarrow$ |
|---|---|---|---|---|---|---|
| 0.05 | Polyp | CRC | 0.952 (0.160) | 0.079 (0.120) | 0.122 (0.120) | 0.515 (0.220) |
| | | AA-CRC | 0.951 (0.160) | 0.070 (0.140) | **0.598 (0.360)** | **0.261 (0.320)** |
| | | CRA (Ours) | 0.951 (0.100) | **0.063 (0.100)**$^\dagger$ | 0.308 (0.340) | 0.406 (0.280) |
| | Crack | CRC | 0.953 (0.080) | 0.051 (0.060) | 0.050 (0.060) | 0.852 (0.160) |
| | | AA-CRC | 0.943 (0.120) | 0.055 (0.100) | **0.110 (0.160)** | **0.686 (0.320)** |
| | | CRA (Ours) | 0.958 (0.060) | **0.040 (0.040)**$^\dagger$ | 0.093 (0.160) | 0.806 (0.260) |
| 0.1 | Polyp | CRC | 0.901 (0.240) | 0.156 (0.200) | 0.326 (0.220) | 0.255 (0.240) |
| | | AA-CRC | 0.901 (0.180) | 0.103 (0.160) | **0.714 (0.360)** | **0.193 (0.300)** |
| | | CRA (Ours) | 0.903 (0.160) | **0.102 (0.120)**$^\dagger$ | 0.548 (0.399) | 0.248 (0.260) |
| | Crack | CRC | 0.903 (0.100) | 0.087 (0.060) | 0.092 (0.080) | 0.519 (0.260) |
| | | AA-CRC | 0.902 (0.100) | 0.069 (0.080) | **0.178 (0.220)** | **0.477 (0.320)** |
| | | CRA (Ours) | 0.906 (0.080) | **0.066 (0.060)**$^\dagger$ | 0.155 (0.220) | 0.551 (0.260) |
| 0.2 | Polyp | CRC | 0.806 (0.340) | 0.262 (0.220) | **0.794 (0.240)** | **0.077 (0.100)** |
| | | AA-CRC | 0.807 (0.240) | 0.160 (0.180) | 0.770 (0.340) | 0.149 (0.240) |
| | | CRA (Ours) | 0.803 (0.200) | **0.138 (0.140)**$^\dagger$ | 0.677 (0.419) | 0.163 (0.180) |
| | Crack | CRC | 0.802 (0.160) | 0.134 (0.080) | 0.335 (0.180) | 0.140 (0.140) |
| | | AA-CRC | 0.807 (0.160) | 0.113 (0.120) | **0.351 (0.200)** | 0.132 (0.080) |
| | | CRA (Ours) | 0.802 (0.120) | **0.089 (0.080)**$^\dagger$ | 0.296 (0.280) | 0.178 (0.120) |

risk control. Precision is calculated as $\frac{1}{|\mathcal{I}_{\text{test}}|} \sum_{i \in \mathcal{I}_{\text{test}}} \frac{|\hat{C}(X_i) \cap Y_i|}{|\hat{C}(X_i)|}$, with higher values indicating more accurate prediction sets. The Normalized Size is given by $\frac{1}{|\mathcal{I}_{\text{test}}|} \sum_{i \in \mathcal{I}_{\text{test}}} \frac{|\hat{C}(X_i)|}{N_i}$, where a lower value signifies a more efficient prediction set.

Figure 2 shows the empirical distributions of coverage rates across both datasets and significance levels $\alpha = 0.1, 0.2$. CRA produces more concentrated distributions centered around the target coverage rate (indicated by the vertical dashed line), with significantly reduced variance compared to CRC and AA-CRC. This confirms that CRA maintains more consistent conditional coverage across diverse inputs.

## 5.2 Experimental Results and Analysis

We evaluate our proposed CRA method against CRC and AA-CRC baselines, presenting quantitative and qualitative results. Table 1 details performance across Polyp and Crack datasets at $\alpha \in \{0.05, 0.1, 0.2\}$, assessing marginal coverage, Coverage Gap (fairness), precision, and prediction set size. For main experiments, we use $K = 5$ strata, as discussed in Section 3.4. A dagger ($\dagger$) denotes statistical significance of CRA's Coverage Gap improvement over the next-best method (one-sided independent samples t-test, $p < 0.05$).

Quantitatively, CRA consistently achieves state-of-the-art conditional risk control, yielding the lowest Coverage Gap across all six settings. This superiority is statistically significant in every case, demonstrating robust equitable coverage without compromising the target marginal coverage. CRA's stratification explicitly addresses conditional coverage, ensuring fairness across diverse subgroups, unlike traditional methods that can exhibit larger disparities. This adaptive approach maintains tight control over coverage probabilities even with heterogeneous data, a common challenge for CRC and AA-CRC. CRA's advantage is more pronounced at higher $\alpha$ levels, where it effectively maintains conditional risk control while producing reasonable prediction set sizes and precision.

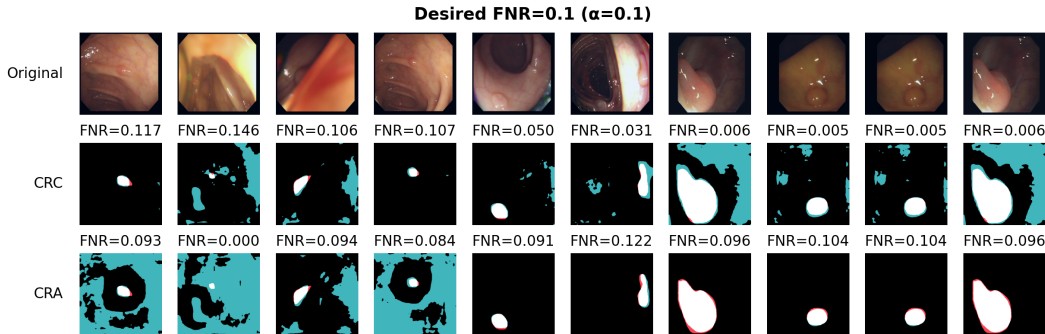

Figure 3: Qualitative comparison of CRC and CRA prediction sets on polyp segmentation at significance level $\alpha = 0.1$. Each column pair shows examples at specific CRC coverage levels (from left to right: 85%, 90%, 95%, 99%, and 99.5%), while CRA coverage remains approximately 90% across all samples. The top row displays original polyp images, the middle row shows CRC prediction sets, and the bottom row presents CRA prediction sets. White pixels indicate true positives, red pixels show false negatives, and teal pixels represent false positives. FNR values (FNR = 1 - coverage) are shown for each prediction.

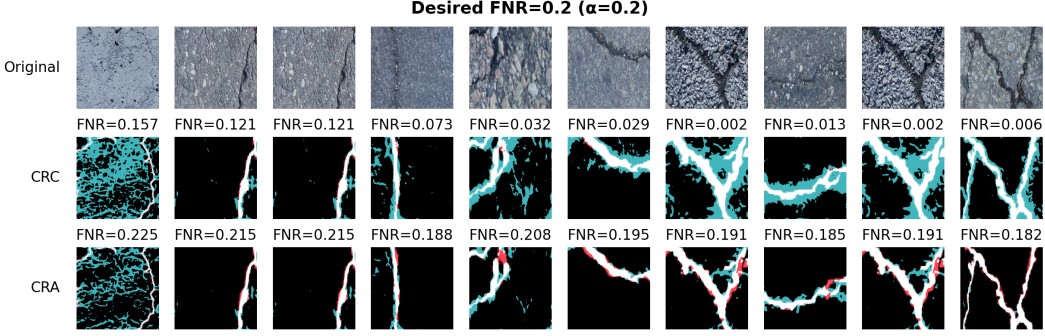

Figure 4: Qualitative comparison of CRC and CRA prediction sets on crack segmentation at significance level $\alpha = 0.2$. Similar to the polyp results, CRA maintains consistent coverage close to the target 80% while CRC shows significant variation. FNR values are shown for each prediction.

Qualitatively, Figures 3 (Polyp, $\alpha = 0.1$) and 4 (Crack, $\alpha = 0.2$) visually illustrate these benefits. While CRC's prediction regions often fluctuate, leading to under- or over-coverage for specific instances, CRA consistently produces prediction regions closer to the target coverage conditionally. This highlights its stability and value for reliable decision support in critical applications where both overall accuracy and fairness are paramount.

Detailed ablation studies on individual components (A.1) and the number of strata $K \in \{2, 3, 4, 5, 6\}$ (A.2), along with a computational complexity analysis (A.3), are provided in the Appendix.

# 6 CONCLUSION

CRA introduces a novel approach to image segmentation, significantly advancing conditional risk control by addressing the critical limitation of existing methods where false negative rates vary dramatically across images. This innovation provides improved conditional risk control for individual cases through a specialized probability calibration framework that enhances pixel-wise reliability and enables group-specific thresholds.

The theoretical foundation of CRA unifies conformal risk control and conformal prediction through weighted quantiles, offering distribution-free valid risk bounds with efficient computation. This enhanced conditional control is paramount for high-stakes applications like medical and engineering

imaging, where reliable, adaptive uncertainty quantification tailored to individual cases is essential for trustworthy AI-assisted decision-making.

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

# A APPENDIX

## A.1 ABLATION STUDY ON INDIVIDUAL COMPONENTS

To understand the contribution of each component in our framework, we conduct comprehensive ablation studies examining individual components. Table 2 and 3 present results for various CRA method configurations, alongside baseline CRC methods, across both Polyp and Crack datasets at significance levels $\alpha = 0.05, 0.1, 0.2$. Table 4 and 5 provide additional insights into the stability of coverage (Coverage Std Mean) and risk control (CVaR Gap).

We evaluate the following CRA variants: **CRA w/o calib+strat** uses only our adaptive score function without probability calibration or stratification (equivalent to CRA (raw, global) in the logs); **CRA w/o calib** incorporates the adaptive score function and stratification but excludes probability calibration (equivalent to CRA (raw, stratified)); **CRA w/o strat** includes the adaptive score function and probability calibration but excludes stratification (equivalent to CRA (calib, global)); and the full **CRA** method combines all components including the adaptive score function, probability calibration, and stratification (equivalent to CRA (calib, stratified)). For comparison, we also include baseline **CRC** (equivalent to CRC (raw, global)) and **CRC w/ strat** (equivalent to CRC (raw, stratified)) methods.

### A.1.1 INTERPRETATION OF RESULTS

The ablation study reveals several key insights into the performance of our proposed CRA framework and the contribution of its individual components:

**Coverage Gap Performance** Across both datasets (Polyp and Crack) and all tested significance levels ($\alpha = 0.05, 0.1, 0.2$), the full **CRA** method (CRA (calib, stratified)) consistently achieves the lowest Coverage Gap. This is a crucial finding, as a lower Coverage Gap indicates better adherence to the target coverage level across different subgroups, which is the primary objective of conditional risk control. The dagger ($\dagger$) symbols in Tables 2 and 3 confirm that this improvement is statistically

Table 2: Complete experiment results comparing all methods across different significance levels ($\alpha$) on the Polyp dataset. We report four performance metrics: Marginal Coverage, Coverage Gap, Precision, and Size. Values are reported as mean (standard deviation). **Bold** values indicate the best performance for each metric. A dagger ($^\dagger$) indicates statistically significant improvement ($p < 0.05$, one-sided t-test) in the metric for the best method compared to the second-best.

(a) Performance comparison of different methods ($\alpha = 0.05$) on Polyp dataset

| Method | Marginal Coverage | Coverage Gap ↓ | Precision ↑ | Size ↓ |
|---|---|---|---|---|
| CRC | 0.952 (0.151) | 0.079 (0.129) | 0.120 (0.110) | 0.516 (0.214) |
| CRC w/ strat | 0.962 (0.123) | 0.068 (0.103) | **0.316 (0.332)** | 0.478 (0.290) |
| CRA w/o calib+strat | 0.953 (0.142) | 0.076 (0.119) | 0.137 (0.182) | 0.562 (0.065) |
| CRA w/o calib | 0.961 (0.122) | 0.068 (0.102) | 0.219 (0.258) | 0.511 (0.272) |
| CRA w/o strat | 0.953 (0.115) | 0.067 (0.093) | 0.255 (0.340) | 0.473 (0.222) |
| CRA | 0.963 (0.107) | **0.062 (0.088)**$^\dagger$ | 0.296 (0.328) | **0.413 (0.272)** |

(b) Performance comparison of different methods ($\alpha = 0.1$) on Polyp dataset

| Method | Marginal Coverage | Coverage Gap ↓ | Precision ↑ | Size ↓ |
|---|---|---|---|---|
| CRC | 0.901 (0.248) | 0.156 (0.193) | 0.318 (0.205) | 0.258 (0.236) |
| CRC w/ strat | 0.909 (0.201) | 0.129 (0.154) | 0.431 (0.386) | 0.284 (0.217) |
| CRA w/o calib+strat | 0.901 (0.211) | 0.143 (0.154) | 0.243 (0.301) | 0.321 (0.066) |
| CRA w/o calib | 0.908 (0.194) | 0.131 (0.143) | 0.338 (0.328) | 0.298 (0.208) |
| CRA w/o strat | 0.904 (0.159) | 0.111 (0.114) | 0.409 (0.415) | 0.307 (0.221) |
| CRA | 0.910 (0.158) | **0.100 (0.122)**$^\dagger$ | **0.554 (0.408)** | **0.247 (0.257)** |

(c) Performance comparison of different methods ($\alpha = 0.2$) on Polyp dataset

| Method | Marginal Coverage | Coverage Gap ↓ | Precision ↑ | Size ↓ |
|---|---|---|---|---|
| CRC | 0.902 (0.212) | 0.198 (0.127) | **0.800 (0.233)** | **0.086 (0.093)** |
| CRC w/ strat | 0.834 (0.263) | 0.203 (0.169) | 0.651 (0.370) | 0.132 (0.122) |
| CRA w/o calib+strat | 0.806 (0.290) | 0.240 (0.163) | 0.391 (0.371) | 0.160 (0.049) |
| CRA w/o calib | 0.813 (0.259) | 0.206 (0.159) | 0.539 (0.384) | 0.149 (0.113) |
| CRA w/o strat | 0.802 (0.208) | 0.158 (0.136) | 0.604 (0.431) | 0.182 (0.185) |
| CRA | 0.812 (0.200) | **0.138 (0.145)**$^\dagger$ | 0.679 (0.415) | 0.164 (0.176) |

significant ($p < 0.05$) when compared to the second-best performing method for Coverage Gap. This demonstrates the synergistic benefits of combining the adaptive score function, probability calibration, and stratification.

**CVaR Gap Performance** The results for CVaR Gap (Conditional Value at Risk Gap), presented in Tables 4 and 5, show a more nuanced picture. For the Polyp dataset at $\alpha = 0.05$, the full **CRA** method achieves the best (lowest) CVaR Gap. However, at higher $\alpha$ values for the Polyp dataset ($\alpha = 0.1$ and $\alpha = 0.2$), **CRA w/o strat** (CRA (calib, global)) and **CRC** (CRC (raw, global)) respectively, show the lowest CVaR Gap. This suggests that while stratification and calibration are generally beneficial for overall coverage adherence, their impact on the most severe coverage shortfalls (captured by CVaR Gap) can vary with the target $\alpha$ and dataset characteristics. For the Crack dataset, the full **CRA** method consistently achieves the lowest CVaR Gap for $\alpha = 0.1$ and $\alpha = 0.2$, and **CRA w/o calib+strat** (CRA (raw, global)) performs best at $\alpha = 0.05$. The statistical significance of these differences is also noted by the dagger symbols.

**Coverage Stability (Coverage Std Mean Per Repeat)** The 'Coverage Std Mean' metric, shown in Tables 4 and 5, quantifies the stability of coverage across different experimental repeats. A lower value indicates more consistent performance. The full **CRA** method (CRA (calib, stratified)) generally demonstrates superior stability, achieving the lowest 'Coverage Std Mean' across most settings. This is particularly evident for the Polyp dataset at $\alpha = 0.05$ and $\alpha = 0.2$, and for the Crack dataset at

Table 3: Performance comparison on the Crack segmentation dataset across different significance levels ($\alpha$).

(a) Performance comparison of different methods ($\alpha = 0.05$) on Crack dataset

| Method | Marginal Coverage | Coverage Gap ↓ | Precision ↑ | Size ↓ |
|---|---|---|---|---|
| CRC | 0.952 (0.073) | 0.051 (0.053) | 0.050 (0.049) | 0.848 (0.170) |
| CRC w/ strat | 0.962 (0.059) | 0.044 (0.042) | 0.063 (0.081) | 0.822 (0.188) |
| CRA w/o calib+strat | 0.952 (0.056) | 0.049 (0.028) | 0.050 (0.047) | 0.842 (0.017) |
| CRA w/o calib | 0.962 (0.059) | 0.044 (0.041) | 0.054 (0.063) | 0.866 (0.125) |
| CRA w/o strat | 0.952 (0.058) | 0.049 (0.031) | 0.066 (0.084) | **0.771 (0.141)** |
| CRA | 0.962 (0.056) | **0.041 (0.040)**$^{\dagger}$ | **0.099 (0.159)** | 0.777 (0.266) |

(b) Performance comparison of different methods ($\alpha = 0.1$) on Crack dataset

| Method | Marginal Coverage | Coverage Gap ↓ | Precision ↑ | Size ↓ |
|---|---|---|---|---|
| CRC | 0.902 (0.110) | 0.088 (0.067) | 0.094 (0.090) | 0.513 (0.254) |
| CRC w/ strat | 0.910 (0.106) | 0.080 (0.071) | 0.113 (0.137) | 0.508 (0.210) |
| CRA w/o calib+strat | 0.903 (0.096) | 0.082 (0.051) | 0.098 (0.102) | 0.430 (0.030) |
| CRA w/o calib | 0.911 (0.098) | 0.077 (0.061) | 0.112 (0.150) | 0.502 (0.166) |
| CRA w/o strat | 0.903 (0.089) | 0.075 (0.047) | 0.144 (0.184) | **0.420 (0.161)** |
| CRA | 0.910 (0.085) | **0.067 (0.054)**$^{\dagger}$ | **0.163 (0.230)** | 0.523 (0.249) |

(c) Performance comparison of different methods ($\alpha = 0.2$) on Crack dataset

| Method | Marginal Coverage | Coverage Gap ↓ | Precision ↑ | Size ↓ |
|---|---|---|---|---|
| CRC | 0.803 (0.152) | 0.132 (0.076) | **0.337 (0.181)** | 0.138 (0.130) |
| CRC w/ strat | 0.808 (0.155) | 0.116 (0.103) | 0.276 (0.196) | 0.134 (0.060) |
| CRA w/o calib+strat | 0.801 (0.131) | 0.099 (0.086) | 0.274 (0.238) | **0.128 (0.013)** |
| CRA w/o calib | 0.809 (0.129) | 0.098 (0.084) | 0.266 (0.231) | 0.135 (0.023) |
| CRA w/o strat | 0.801 (0.123) | 0.100 (0.072) | 0.333 (0.268) | 0.151 (0.119) |
| CRA | 0.809 (0.115) | **0.090 (0.073)**$^{\dagger}$ | 0.298 (0.277) | 0.176 (0.122) |

$\alpha = 0.1$ and $\alpha = 0.2$, where its performance is statistically significantly better than the next best method. This highlights the robustness of the full CRA approach in maintaining consistent coverage guarantees.

**Precision and Size**  In terms of Precision and Size (Tables 2 and 3), the full **CRA** method often achieves high precision, indicating that a large proportion of its predictions are correct. While it might not always yield the smallest prediction sets (Size), it strikes a good balance between controlling coverage and providing informative, precise sets. For instance, on the Polyp dataset at $\alpha = 0.1$, CRA achieves the highest precision and the smallest size. On the Crack dataset, CRA also achieves the highest precision at $\alpha = 0.05$ and $\alpha = 0.1$.

**Role of Individual Components**

- **Stratification:** Comparing CRA w/o calib (CRA (raw, stratified)) with CRA w/o calib+strat (CRA (raw, global)), and CRA (CRA (calib, stratified)) with CRA w/o strat (CRA (calib, global)), we observe that stratification generally leads to a lower Coverage Gap and often better CVaR Gap, especially at lower $\alpha$ values. This underscores the importance of addressing subgroup-specific coverage requirements.

- **Calibration:** Comparing CRA w/o strat (CRA (calib, global)) with CRA w/o calib+strat (CRA (raw, global)), and CRA (CRA (calib, stratified)) with CRA w/o calib (CRA (raw, stratified)), probability calibration consistently improves precision and often helps in reducing the Coverage Gap and CVaR Gap. This suggests that well-calibrated probabilities are crucial for constructing reliable prediction sets.

Table 4: Additional performance metrics: Coverage Standard Deviation Mean per Repeat and CVaR Gap. Values are reported as mean (standard deviation). **Bold** values indicate the best performance for each metric. A dagger ($\dagger$) indicates statistically significant improvement ($p < 0.05$, one-sided t-test) in the metric for the best method compared to the second-best.

(a) Additional metrics ($\alpha = 0.05$) on Polyp dataset

| Method | Coverage Std Mean ↓ | CVaR Gap ↓ |
|---|---|---|
| CRC | 0.149 (0.151) | 0.583 (0.219) |
| CRC w/ strat | 0.120 (0.123) | 0.442 (0.239) |
| CRA w/o calib+strat | 0.139 (0.142) | 0.532 (0.223) |
| CRA w/o calib | 0.119 (0.122) | 0.431 (0.249) |
| CRA w/o strat | 0.113 (0.115) | 0.414 (0.190) |
| CRA | **0.105 (0.107)**$^\dagger$ | **0.374 (0.217)**$^\dagger$ |

(b) Additional metrics ($\alpha = 0.1$) on Polyp dataset

| Method | Coverage Std Mean ↓ | CVaR Gap ↓ |
|---|---|---|
| CRC | 0.247 (0.248) | 0.694 (0.211) |
| CRC w/ strat | 0.199 (0.201) | 0.520 (0.245) |
| CRA w/o calib+strat | 0.209 (0.211) | 0.547 (0.214) |
| CRA w/o calib | 0.192 (0.194) | 0.487 (0.238) |
| CRA w/o strat | 0.157 (0.159) | **0.379 (0.201)**$^\dagger$ |
| CRA | 0.156 (0.158) | 0.389 (0.216) |

(c) Additional metrics ($\alpha = 0.2$) on Polyp dataset

| Method | Coverage Std Mean ↓ | CVaR Gap ↓ |
|---|---|---|
| CRC | 0.344 (0.212) | **0.240 (0.289)**$^\dagger$ |
| CRC w/ strat | 0.291 (0.263) | 0.418 (0.266) |
| CRA w/o calib+strat | 0.290 (0.290) | 0.507 (0.185) |
| CRA w/o calib | 0.258 (0.259) | 0.426 (0.223) |
| CRA w/o strat | 0.208 (0.208) | 0.321 (0.200) |
| CRA | **0.199 (0.200)**$^\dagger$ | 0.305 (0.229) |

**Conclusion from Ablation** The ablation study confirms that all components of the full CRA method contribute to its overall superior performance, particularly in achieving low Coverage Gap and stable coverage. While there can be specific trade-offs for CVaR Gap at certain $\alpha$ levels, the comprehensive benefits of combining adaptive scores, probability calibration, and stratification make the full CRA method a robust solution for conditional risk control in segmentation tasks.

A.2 ABLATION STUDY ON NUMBER OF STRATA K

We investigate the impact of the number of strata $K$ on CRA's performance at $\alpha = 0.1$. Note that $K = 1$ would effectively degenerate CRA to a method without stratification. Table 6 presents results for $K \in \{2, 3, 4, 5, 6\}$ on both Polyp and Crack datasets.

As $K$ increases, the Coverage Gap generally decreases, indicating improved conditional risk control. However, this comes at the cost of a slight increase in prediction set Size and a slight decrease in Precision. The overall performance changes are minor across this range of $K$, suggesting CRA is robust to its exact choice. While higher $K$ can reduce the Coverage Gap, $K$ cannot be excessively large due to the limited amount of calibration data available for robust quantile estimation within each stratum. Our choice of $K = 5$ for main experiments offers a good balance.

Table 5: Additional performance metrics: Coverage Standard Deviation Mean per Repeat and CVaR Gap on Crack dataset.

(a) Additional metrics ($\alpha = 0.05$) on Crack dataset

| Method | Coverage Std Mean ↓ | CVaR Gap ↓ |
|---|---|---|
| CRC | 0.072 (0.073) | 0.186 (0.168) |
| CRC w/ strat | 0.058 (0.059) | 0.166 (0.114) |
| CRA w/o calib+strat | 0.056 (0.056) | **0.131 (0.037)**[†] |
| CRA w/o calib | 0.058 (0.059) | 0.159 (0.120) |
| CRA w/o strat | 0.058 (0.058) | 0.144 (0.056) |
| CRA | **0.055 (0.056)** | 0.153 (0.117) |

(b) Additional metrics ($\alpha = 0.1$) on Crack dataset

| Method | Coverage Std Mean ↓ | CVaR Gap ↓ |
|---|---|---|
| CRC | 0.110 (0.110) | 0.190 (0.154) |
| CRC w/ strat | 0.105 (0.106) | 0.206 (0.153) |
| CRA w/o calib+strat | 0.096 (0.096) | 0.168 (0.093) |
| CRA w/o calib | 0.097 (0.098) | 0.182 (0.126) |
| CRA w/o strat | 0.089 (0.089) | 0.156 (0.079) |
| CRA | **0.085 (0.085)**[†] | **0.156 (0.105)**[†] |

(c) Additional metrics ($\alpha = 0.2$) on Crack dataset

| Method | Coverage Std Mean ↓ | CVaR Gap ↓ |
|---|---|---|
| CRC | 0.159 (0.152) | 0.186 (0.094) |
| CRC w/ strat | 0.154 (0.155) | 0.206 (0.161) |
| CRA w/o calib+strat | 0.131 (0.131) | 0.179 (0.122) |
| CRA w/o calib | 0.129 (0.129) | 0.166 (0.120) |
| CRA w/o strat | 0.123 (0.123) | 0.166 (0.081) |
| CRA | **0.115 (0.115)**[†] | **0.148 (0.096)**[†] |

Table 6: Ablation study of CRA performance with varying number of strata $K$ at $\alpha = 0.1$. We report Marginal Coverage (target is $1 - \alpha$), Coverage Gap (lower is better ↓), Precision (higher is better ↑), and normalized Size (lower is better ↓). Values are mean (std $\times \sqrt{399}$).

| Dataset | K | Marginal Coverage | Coverage Gap ↓ | Precision ↑ | Size ↓ |
|---|---|---|---|---|---|
| Polyp | 2 | 0.902 (0.180) | 0.115 (0.140) | **0.550 (0.419)** | **0.240 (0.280)** |
| | 3 | 0.902 (0.160) | 0.110 (0.140) | 0.548 (0.419) | 0.243 (0.280) |
| | 4 | 0.902 (0.160) | 0.108 (0.120) | 0.545 (0.399) | 0.244 (0.260) |
| | 5 | 0.903 (0.160) | 0.102 (0.120) | 0.540 (0.399) | 0.248 (0.260) |
| | 6 | 0.903 (0.160) | **0.098 (0.120)** | 0.535 (0.399) | 0.252 (0.260) |
| Crack | 2 | 0.905 (0.100) | 0.075 (0.080) | **0.160 (0.240)** | **0.540 (0.280)** |
| | 3 | 0.905 (0.100) | 0.072 (0.080) | 0.159 (0.240) | 0.543 (0.280) |
| | 4 | 0.906 (0.080) | 0.070 (0.060) | 0.158 (0.220) | 0.545 (0.260) |
| | 5 | 0.906 (0.080) | 0.066 (0.060) | 0.155 (0.220) | 0.551 (0.260) |
| | 6 | 0.906 (0.080) | **0.063 (0.060)** | 0.152 (0.220) | 0.555 (0.260) |

## A.3 COMPUTATIONAL COMPLEXITY ANALYSIS

Table 7 shows 10-run averages on Intel i9-13900K with RTX 4090. CRA achieves over $27\times$ speedup versus CRC and $993\times$ versus AA-CRC due to neural network training overhead. Our weighted quantile algorithm from Section 3.1 provides exact precision, unlike CRC's grid-dependent approximation.

Table 7: Runtime Comparison (10 Runs) in Seconds

| CRC | AA-CRC | CRA (Ours) |
|---|---|---|
| $13.21 \pm 0.13$ | $477.20 \pm 51.33$ | $\mathbf{0.48 \pm 0.01}$ |

LLM ASSISTANCE

Claude-Opus-4.1, a large language model, served as a general-purpose assistant during manuscript preparation. It refined paper sections for clarity and grammar. Authors maintained full responsibility for reviewing and validating all LLM-assisted content, ensuring accuracy and scientific standards. Claude-Opus-4.1 was not involved in core research, experimental design, data collection, or primary analysis. All scientific content, conclusions, and errors remain solely the authors' responsibility.

