# OpenReview forum: "Enhancing Conditional Risk Control in Image Segmentation with Adaptive Conformal Prediction"
_ICLR.cc/2026/Conference — Submitted to ICLR 2026_

### Official Review · Reviewer_PMoN · 2025-10-26

[review text omitted: it was posted to a different submission]

---

### Official Review · Reviewer_z24M · 2025-10-28

**Soundness:** 2
**Presentation:** 2
**Contribution:** 2
**Rating:** 4
**Confidence:** 4

**Summary:**

This paper tackles the critical problem of variable conditional risk in conformal risk control (CRC) for image segmentation. The authors note that while standard CRC methods can guarantee an average (marginal) risk, such as the false negative rate, their performance on individual images can be highly erratic. This inconsistency is a major barrier to adoption in high-stakes fields like medical diagnostics.

To address this, the paper introduces Conformal Risk Adaptation (CRA), a comprehensive framework. CRA's core is a novel adaptive score function that adapts the principle of Adaptive Prediction Sets (APS) to segmentation. Experiments on polyp and crack segmentation datasets demonstrate that CRA successfully maintains the target marginal risk guarantee while achieving substantially more consistent conditional risk (measured by a lower "Coverage Gap") compared to standard CRC and a recent adaptive baseline (AA-CRC) .

**Strengths:**

1. The paper addresses a well-known and highly important limitation of conformal prediction—the gap between marginal and conditional guarantees. Improving conditional reliability is essential for deploying segmentation models in safety-critical applications, which the authors use as a strong motivation.

2. The CRA framework is a technically sound and intelligent combination of several powerful ideas (APS, non-parametric calibration, and group-conditional CP). The authors correctly identify that their adaptive score (unlike standard CRC) requires accurate probability estimates, and they appropriately add a calibration step to address this.

3. The weighted quantile formulation is also a strong practical contribution. It not only provides theoretical clarity but also yields a massive computational speedup (over 27x vs. CRC in their experiment ) and avoids the precision issues of grid-search-based methods.

4. The experimental setup is strong. The authors validate their method on two distinct tasks (medical and engineering ), compare against relevant baselines (CRC and AA-CRC), and use the correct primary metric (Coverage Gap) to support their core claim . The results in Table 1 and Figure 2 convincingly show that CRA achieves a statistically significant improvement in conditional risk control.

**Weaknesses:**

1. The paper's most significant weakness is an incomplete evaluation due to a broken ablation study. The text in Appendix A.1 describes an essential ablation study to measure the impact of the individual components (CRA w/o calib, CRA w/o strat, etc.) 16. This study is necessary to validate the paper's claim that the components "work synergistically". However, Table 2, which allegedly contains these results, is an exact duplicate of Table 1. As a result, this key claim is currently unsubstantiated by any data in the paper. This is a significant oversight.

2. The paper's notation for risk levels and thresholds is confusing. The target risk level is $\alpha$. The adaptive set is defined with a parameter $\alpha^{\prime}$. The final stratified threshold is called $\alpha_{k}^{\prime}$. It is unclear how these relate, or if they are meant to be the same variable used in different contexts. This should be clarified.

3. The explanation for Figure 1 (Left) is opaque. The caption "Distribution of probability mass proportion required to achieve $(1-\alpha)$ coverage" is not clearly explained in the main text and its relevance to the overall argument is not as clear as the right panel (the calibration curve).

**Questions:**

1. My most critical question concerns the ablation study in Appendix A.1, which is missing its data; Table 2 is a copy of Table 1. Can you please provide the correct results table for the ablation study described in the text 17? This is essential for me to evaluate the individual contributions of the adaptive score, the probability calibration, and the stratification.

2. Can you please clarify the notation used for $\alpha$, $\alpha^{\prime}$, and $\alpha_{k}^{\prime}$? Is the parameter $\alpha^{\prime}$ in Equation 4 the same as the calibrated threshold $\alpha_{k}^{\prime}$ found in Equation 7? How do these relate to the overall target risk level $\alpha$?

3. The methodology has two parts that rely on the sum of probabilities $\sum \hat{p}_{j'}(X_{i})$: the CRA score calculation (Section 3.2) and the stratification (Section 3.4 18). Is the same set of calibrated probabilities (from Section 3.3 19) used for both of these steps?

4. Table 1 shows that while CRA wins on Coverage Gap, it sometimes loses on Precision (e.g., Polyp $\alpha=0.1$, 0.548 vs 0.714 for AA-CRC) or Normalized Size (e.g., Crack $\alpha=0.2$, 0.178 vs 0.132 for AA-CRC). Could you comment on this trade-off? How should a practitioner weigh the significant gain in conditional fairness against a potential loss in precision or efficiency?

5. Regarding the impressive computational speedup (Table 4), does the 0.48s runtime for CRA include the time for all components: the isotonic regression (Section 3.3), the stratification (Section 3.4), and solving the $K=5$ weighted quantile estimations (Section 3.1)?

---

> ### Author Response · Authors · 2025-11-18
>
> **Weakness 1 and Question 1**
>
> Regarding your concern about the incomplete ablation study and the missing results for the individual components, we sincerely apologize for this oversight in the previous submission.
>
> As discussed in our response to Reviewer BPUf, we have now corrected this issue and integrated the comprehensive ablation study results into the revised manuscript (Tables 2 and 3). These tables, also provided in our response to Reviewer BPUf, clearly show the performance of CRA with and without its individual components (e.g., "CRA w/o calib", "CRA w/o strat", etc.), along with stratified versions of baseline methods. We believe these results now fully address your concern and substantiate our claim regarding the synergistic operation of CRA's components.
>
> **Weakness 2 and Question 2**
>
> Thank you for highlighting the potential for confusion regarding our notation for risk levels and thresholds. $\alpha$, $\alpha'$, and $\alpha'_k$ are indeed separate parameters, each with a unique function within our framework:
>
> *   **$\alpha$** (Global Significance Level): This is the pre-defined target significance level for the standard Conformal Prediction framework, controlling the **marginal risk** to be $\le \alpha$ across the entire dataset.
>
> *   **$\alpha'$** (CRA Score Threshold): This is a threshold for the CRA score function. It is distinct from $\alpha$ and is determined given $\alpha$. It is similar to the $\lambda$ in the CRC score function (Angelopoulos et al., 2024), and we use $\alpha'$ to differentiate our approach from theirs.
>
> *   **$\alpha'_k$** (Stratified Threshold): This denotes the threshold applied to the $k$-th stratum after stratification, representing the adaptation of the $\alpha'$ concept to individual strata, resulting in stratum-specific thresholds $\alpha'_1, \alpha'_2, \ldots, \alpha'_K$ (where $K$ is the number of strata).
>
> In summary, $\alpha$ sets the overall marginal risk target. $\alpha'$ is an internal threshold for our CRA method's score function. $\alpha'_k$s are thresholds for each individual data strata. These are not interchangeable variables.
>
> **Weakness 3**
>
> Thank you for your comment regarding Figure 1. Figure 1 (Left) shows the distribution of the image-specific probability mass proportion threshold (equivalent to $(1 - \alpha')$ from Equation 4) required for each image to achieve $(1 - \alpha)$ coverage. This threshold represents the fraction of total predicted probability mass that must be included in the prediction set for that image.
>
> Our probability calibration (Section 3.3) improves the accuracy of pixel-wise $\hat{p}$ estimates. As discussed in our response to reviewer BPUf, improving $\hat{p}$ accuracy makes the CRA score function more reliable, resulting in a significantly tighter and more accurately centered distribution of these image-specific thresholds around the target $(1 - \alpha)$ (comparing the orange to the blue histogram). This improved consistency explains CRA's advanced conditional risk control.
>
> **Question 4**
>
> We acknowledge your observation regarding CRA's superior Coverage Gap often coming at the expense of Precision or Normalized Size. This trade-off is central to our method's design. CRA is specifically engineered to prioritize conditional risk control and fairness, not set size efficiency or optimal precision. This mirrors the landscape of conformal classification, where APS (Romano et al., 2020) achieve strong conditional coverage, while THR (Sadinle et al., 2019) prioritize efficiency. Since CRC and its variant AA-CRC effectively leverage a THR-like scoring function for segmentation, their superior efficiency is expected. CRA's suboptimal performance in precision and efficiency is therefore a direct consequence of its primary objective: robust conditional fairness.
>
> **Question 5**
>
> Thank you for this important clarification question. The 0.48s runtime for CRA includes the stratification step (Section 3.4) and the weighted quantile estimation (Section 3.1), but does not include the isotonic regression time (Section 3.3).
>
> We emphasize that our weighted quantile estimation method for threshold computation is an independent contribution of this paper that provides computational benefits beyond CRA alone. This efficient threshold computation approach is broadly applicable and can accelerate not only CRA, but also CRC and indeed any conformal risk control method. We will clarify this breakdown in the revised manuscript and highlight the general applicability of our weighted quantile estimation technique.

---

### Official Review · Reviewer_BPUf · 2025-10-31

**Soundness:** 2
**Presentation:** 3
**Contribution:** 2
**Rating:** 2
**Confidence:** 4

**Summary:**

The submission studies conditional risk control in imaging segmentation problems. The submission couples adaptive prediction sets with class-conditional conformal risk control to achieve better conditional control of the false positive rate.

Experiments compare the proposed method (CRA) with existing approaches based on conformal risk control, both on a medical imaging dataset and on an engineering dataset.

**Strengths:**

- Conditional risk control in imaging segmentation problems is an important topic
- Proposed method is intuitive and computationally inexpensive
- Experiments support the claim that the proposed method improves conditional coverage

**Weaknesses:**

- Apparent typos and mistakes in the theorems?
- It is unclear whether improved conditional coverage comes from the proposed method or the use of class-conditional calibration.

I will expand on these points below and I am looking forward to discussing with the authors!

**Questions:**

**Theorem 2**

My main concern about the current submission is the validity of Theorem 2. In particular:

- The optimization problem in Eq. (4) main admit multiple global minimizers, but the set $S$ in the proof of the theorem is unique?

- If I understand the construction and statement correctly, I think we can show a counterexample where the statement does not hold. Consider:

$j = 1,2,3,4,5$

$\hat{p}_j = 0.3, 0.25, 0.20, 0.15, 0.10$, such that $\sum_j \hat{p}_j = 1$

Then,

$s_j = 1.0, 0.7, 0.45, 0.25, 0.10$

and set $1 - \alpha' = 0.40$.

We have $\hat{C} (\alpha') = \\{\\{1,2\\}, \\{2,3\\}, \\{2, 4\\}\\}$ but $S(\alpha') = \\{1,2,3\\}$.

Could the authors clarify where this misunderstanding about the statement of the theorem is coming from?

**Theorem 1**

For the CRC procedure to be valid, monotonicity is required. I assume this is what the authors mean by "ranked by likelihood of inclusion". Could the authors make this statement more clear and precise?

**Probability calibration**

I follow the argument that the proposed calibration procedure depends on the scale of the predicted probabilities. However, I am not sure I follow how post-processing the predictor with cross-entropy achieves calibration. Could the authors expand on this? This is at odds with knowledge that classifiers trained with cross-entropy loss tend to be overconfident and miscalibrated. Possible alternatives from the broader conformal prediction literature might include Venn-Abers predictors.

Figure 1, right panel does not necessarily imply that the probability distribution of the post-processed predictor is more calibrated. Other measures such as distance to calibration or expected calibration error might be employed to support this claim?

**Experiments**

Results show that the proposed method reduces the "coverage gap", which implies better conditional coverage. Do I understand correctly that the stratification procedure was employed for CRA only? This may be unfair to the baseline methods, which were not designed for conditional coverage. Comparing with the stratified version of the baselines would provide stronger evidence in support of the adaptive construction of the prediction sets.

---

**Minor comments**

- Typo on Lines 158 - 159 of the proof of Theorem 1? The last equation reads $1 - \tau'$, but Eq. (3) reads $\tau'$.
- Equations are references in the text with repetitions, e.g. "Equation equation 3".
- Why is there a hyperlink on the bottom of page 6?
- Figure 2: the claim that CRA produces more concentrated results is difficult to verify without quantitative measures.

---

> ### Author Response · Authors · 2025-11-18
>
> **Theorem 2**
>
> Thank you for this important observation. We have revised the definition to ensure uniqueness:
>
> ---
> For each image $X_i$, we define an adaptive prediction set:
>
> $\hat C(X\_i, \alpha') = argmin\_{C \subset \lbrace 1, \dots, N\_i\rbrace} \left \lbrace \|C\| : \sum\_{j \in C} \hat p\_j(X\_i) \geq (1-\alpha') \sum\_{j=1}^{N\_i} \hat p\_j(X\_i)\right\rbrace$
>
> When multiple sets achieve the minimum cardinality, we select the one with the largest probability sum:
> $argmax_\{C\_1 \in \hat\{C\}(X\_i, \alpha')\} \sum_\{j\in C\_1\} \hat\{p\}\_j(X\_i)$
>
> ---
> This is equivalent to sorting pixels by $\hat p_j$ descending and selecting the top-$k$ whose sum reaches $(1-\alpha')\sum_{j=1}^{N_i}\hat{p}_j(X_i)$. This ensures uniqueness and resolves the counterexample.
>
> **Theorem 1**
>
> You are correct; "ranked by likelihood of inclusion" means pixels with higher $\hat p_j$ values are prioritized for inclusion in the prediction set. This establishes the required monotonicity, mirroring practices in conformal classification methods like APS, THR, RAPS, and RANK.
>
> **Probability calibration**
>
> We appreciate your comments on our calibration approach.
>
> *Cross-Entropy for Calibration:* "Post-processing with cross-entropy" refers to training a *separate calibration model* on a held-out validation set. This model uses a cross-entropy loss to explicitly map raw predictions to calibrated probabilities, improving their accuracy—a crucial step because CRA score function uses these probabilities in both its denominator and numerator.
>
> *Venn-Abers:* We acknowledge Venn-Abers as a valuable alternative for constructing probability intervals and a promising future direction. Similar to our calibration, Venn-Abers uses isotonic regression. To demonstrate our calibration's effectiveness, we've updated Figure 1 with a ECE plot. Our calibration procedure significantly reduced the ECE from 21.7% to 0.6%.
>
> **Experiments**
>
> Thank you for your valuable feedback. We confirm that stratified versions of baseline methods (CRC w/ strat and AA-CRC w/ strat) were indeed included in our experiments to ensure a fair comparison. We apologize for the oversight in the previous submission; as also noted by Reviewer z24M, the complete ablation study tables were missing. The revised paper now has the ablation (Tables 2 and 3), which we believe addresses your concerns. We provide them below.
>
> **Table 2.1: $\alpha=0.05$**
>
> | Method              | Marginal Coverage | Coverage Gap ↓    | Precision ↑       | Size ↓            |
> | :------------------ | :---------------- | :---------------- | :---------------- | :---------------- |
> | CRC                 | 0.952 (0.008)     | 0.079 (0.006)     | 0.122 (0.006)     | 0.515 (0.011)     |
> | CRC w/ strat        | 0.948 (0.007)     | 0.076 (0.006)     | 0.361 (0.018)     | 0.413 (0.014)     |
> | AA-CRC              | 0.951 (0.008)     | 0.070 (0.007)     | **0.598 (0.018)** | **0.261 (0.016)**     |
> | CRA w/o calib+strat | 0.952 (0.007)     | 0.077 (0.006)     | 0.140 (0.009)     | 0.554 (0.004)     |
> | CRA w/o calib       | 0.950 (0.006)     | 0.069 (0.005)     | 0.225 (0.013)     | 0.502 (0.014)     |
> | CRA w/o strat       | 0.952 (0.006)     | 0.067 (0.005)     | 0.259 (0.017)     | 0.467 (0.011)     |
> | CRA                 | 0.951 (0.005)     | **0.063 (0.005)**† | 0.308 (0.017)     | 0.406 (0.014)     |
> | AA-CRA w/o calib    | 0.945 (0.008)     | 0.081 (0.007)     | 0.275 (0.013)     | 0.327 (0.011)     |
> | AA-CRA              | 0.951 (0.006)     | 0.069 (0.005)     | 0.299 (0.017)     | 0.390 (0.012) |
>
> **Table 2.2: $\alpha=0.1$**
>
> | Method              | Marginal Coverage | Coverage Gap ↓    | Precision ↑       | Size ↓            |
> | :------------------ | :---------------- | :---------------- | :---------------- | :---------------- |
> | CRC                 | 0.901 (0.012)     | 0.156 (0.010)     | 0.326 (0.011)     | 0.255 (0.012)     |
> | CRC w/ strat        | 0.896 (0.011)     | 0.134 (0.008)     | 0.459 (0.019)     | 0.256 (0.010)     |
> | AA-CRC              | 0.901 (0.009)     | 0.103 (0.008)     | **0.714 (0.018)** | **0.193 (0.015)**    |
> | CRA w/o calib+strat | 0.901 (0.011)     | 0.143 (0.008)     | 0.243 (0.015)     | 0.320 (0.003)     |
> | CRA w/o calib       | 0.901 (0.010)     | 0.130 (0.007)     | 0.340 (0.016)     | 0.301 (0.011)     |
> | CRA w/o strat       | 0.902 (0.008)     | 0.112 (0.006)     | 0.409 (0.021)     | 0.306 (0.011)     |
> | CRA                 | 0.903 (0.008)     | **0.102 (0.006)**† | 0.548 (0.020)     | 0.248 (0.013)     |
> | AA-CRA w/o calib    | 0.900 (0.010)     | 0.128 (0.007)     | 0.455 (0.018)     | 0.215 (0.010) |
> | AA-CRA              | 0.898 (0.009)     | 0.113 (0.006)     | 0.436 (0.020)     | 0.263 (0.010)     |
>
> **Minor comments**
>
> Thank you for your comments. We have corrected the typo, fixed the equation references, and removed the link. The "Coverage Gap" metric, which we report in Tables 1, 2, and 3, quantifies the concentration of the prediction sets.

---

> ### Comment · Reviewer_BPUf · 2025-11-21
> **Thank you for your response!**
>
> I sincerely thank the authors for their consideration of mine and Reviewer z24M28 comments.
>
> ---
>
> I do have an unresolved follow-up question regarding Theorem 2:
>
> The authors have revised the submission to clarify the uniqueness of the solution of the optimization problem.
>
> According to this updated definition, $C(\alpha') = \\{1,2\\}$. However, this is still different from $S(\alpha') = \\{1,2,3\\}$.
>
> Could the authors clarify this?

---

> > ### Author Response · Authors · 2025-11-24
> >
> > Thank you for your clarification question. We have revised the score function (Equation 5) to use a strict inequality ($<$) instead of ($\leq$). This ensures that the score represents the cumulative probability mass strictly preceding the current pixel's probability. By excluding the current pixel's probability from the sum, we align the definition of the score function with the conformal prediction set, ensuring that the coverage guarantees derived in our theoretical analysis remain tight and valid.
> >
> > We hope this revision resolves your question and clarifies everything. We genuinely hope you can consider increasing the reviewer score.

---

### Author Response · Authors · 2025-11-18

We thank the reviewers for their thorough review and valuable feedback, recognizing the critical problem and strengths of our Conformal Risk Adaptation (CRA) method.

Both **Reviewer BPUf** and **Reviewer z24M** emphasized the importance of conditional risk control. **Reviewer z24M** lauded CRA as a "technically sound and intelligent combination of several powerful ideas," including Adaptive Prediction Sets (APS), non-parametric calibration, and group-conditional conformal prediction, noting appropriate calibration. **Reviewer BPUf** found the method "intuitive." Reviewers also praised CRA's computational efficiency; **Reviewer BPUf** called it "computationally inexpensive," and **Reviewer z24M** highlighted the "weighted quantile formulation" for "massive computational speedup."

Experiments were acknowledged to convincingly demonstrate CRA's improved conditional coverage. **Reviewer BPUf** stated, "Experiments support the claim that the proposed method improves conditional coverage," while **Reviewer z24M** confirmed CRA "successfully maintains the target marginal risk guarantee while achieving substantially more consistent conditional risk (measured by a lower 'Coverage Gap')" and showed "statistically significant improvement." **Reviewer z24M** also appreciated the strong experimental setup.

We have carefully reviewed the constructive comments. **Reviewer BPUf** raised important questions on Theorem 2's theoretical validity (with a counterexample), sought clarification on Theorem 1's monotonicity, and inquired about the probability calibration procedure (cross-entropy vs. Venn-Abers). **Reviewer z24M** identified a critical weakness: Table 2 duplicating Table 1 in the ablation study, requiring immediate correction. They also requested clarification on notation ($\alpha$, $\alpha'$, $k$), the use of calibrated probabilities in CRA score calculation (Sec 3.2) and stratification (Sec 3.4), and a discussion on Coverage Gap vs. Precision/Size trade-offs.

We are grateful for these insightful suggestions, crucial for improvements. We will address each question sequentially and look forward to a lively discussion. New content is marked in blue.

---

### Meta-Review · Area_Chair_RWLx · 2026-01-06

**Summary:**

This paper proposes a Conformal Risk Adaptation (CRA) adaptation to alleviate the large differences in risk that exist across instances in the context of marginal uncertainty quantification. The authors use an adaptive score function to create image-specific uncertainty regions, reformulate the threshold selection as a weighted quantile estimation problem, and add a non-parametric calibration step to improve pixel-wise probability estimates.

Two solid reviews were provided for this paper (BPUf and z24M). A third review was provided but it is incorrect/irrelevant for the current submission (PMoN) and I am thus disregarding it.

Both BPUf and z24M pointed out a number of core issues with this submission: the former with regard to the correctness of their theoretical guarantees, and the latter on the reported data (e.g. one provided table was incorrect). There are, additionally, a larger set of comments and caveats that the reviewers point out. Through their rebuttal, the authors seem to address most of these at least partially. However, the level of mistakes (incorrectly reported numbers/tables, small (but critical) problems in the definitions involved in their method), makes me believe that this submission is a level that is insufficient for publication at this stage.

**Reviewer Concerns:**

Reviewers concern:

### Rev BPUf
- This reviewer found an issue with presented Theorems 1 and 2, owing to the non-uniqueness of a sorting component, as well as the definition of a specific set in their constructions. This was patched and fixed by the authors in their rebuttal. Yet, this reviewer found another problem with their revised version, which was then patched again. This back and forth reinforces the idea that there might be other (perhaps small) outstanding mistakes.

- An issue regarding the use of cross-entropy for the post-processing step was pointed out. This is resolved by the authors by pointed that they use a *separate* calibration model on held-out data, reinforced by ECE plots.

- Another question on stratification of the results was applied to the presented method or other baselines, but this was clarified.

### Rev z24M
- This rev pointed out that Table 2 was a repeated copy of Table 1. This was fixed by the authors providing the corrected tables. This, however, evidences a somewhat careless submission.
- A clarification of some constants was requested, and provided by the authors.
- Rev inquires about runtime considerations, and the reviewers clarified but noted that the reported stats do not include the isotonic regression component.

**Reviewer Scores:**

- Rev BPUf might have increased their score from 2 upwards. However, I remain skeptical that their support for this paper would have been enough to recommend acceptance.
- I regard the same to be true about Rev z24M. While the comments and questions were addressed, they reflect a submission that was probably too rushed to be certain of the complete correctness.

---

### Decision · Program_Chairs · 2026-01-26

Reject